# Hydrochemical Characteristics of Mine Water and Their Significance for the Site Selection of an Underground Reservoir in the Shendong Coal Mining Area

Yangnan Guo [1], Guoqing Li [2,3,*], Lei Wang [2,3] and Zheng Zhang [2,3]

1   China Energy Group, Shendong Coal Technology Research Institute, Yulin 719315, China
2   School of Earth Resources, China University of Geosciences, Wuhan 430074, China
3   State Key Laboratory of Water Resource Protection and Utilization in Coal Mining, Beijing 102211, China
*   Correspondence: ligq@cug.edu.cn

**Abstract:** Underground reservoir technology can mitigate water shortage and pollution problems in water shortage coal mining areas and has a good application prospect. While still a new technology, the theory and method of underground reservoirs need to be improved. This research focused on the hydrochemical characteristics of mine water and their significance for the site selection of underground reservoirs. With the Shendong coal mining area as a case study, the hydrochemical major ions, toxicological indexes, and stable isotopes of hydrogen and oxygen were tested for the mine water samples, and the water quality was quantitatively evaluated and the origins of over-limit variables were investigated by hydrogeochemical numerical simulation and ionic ratio analysis. The influencing factors of water quality were analyzed and the significance of mine water quality for the site selection of underground reservoirs was discussed. The results show that the main over-standard variables are $Na^+$, $F^-$, $SO_4^{2-}$, TDS, and sodium ion adsorption ratio (SAR), and a strong positive correlation exists between $F^-$ and SAR and a negative correlation exists between $F^-$ and $Ca^{2+}$. $Na^+$ in mine water originates from the dissolution of halite and silicate rocks, as well as reverse cation exchange. $F^-$ originates from reverse cation exchange and the displacement between $OH^-$ in alkaline water and $F^-$ adsorbed on the surface of minerals. On the whole, the mine water quality is better on the east than on the west of the WL River. The water–rock interactions in goaf increase the concentrations of $F^-$ and $Ca^{2+}$ and SAR. The areas where the mine water samples have low concentrations of $Na^+$, $F^-$, and low SAR values, such as the shallow coal seams at the SGT, DLT, and WL mines, are favorable sites for the underground reservoir. The outcomes may benefit the reasonable site selection of underground reservoirs in similar coal mining areas with water shortage.

**Keywords:** underground reservoir; site selection; water quality; hydrochemistry; stratigraphic texture

## 1. Introduction

China is a country that experiences water shortages, with a per capita water resource of approximately 2214 $m^3$, only a quarter of the world average. Underground coal mining is often accompanied by the discharge of a large amount of mine water. On average, 1.87 $m^3$ of mine water will be discharged for one ton of coal produced, and the total amount of mine water is approximately $6.88 \times 10^9$ $m^3$ per year in China [1]. Therefore, the reuse of mine water is of great significance and has a good prospect. The Shendong coal mining area is located in the arid and semi-arid region of Northwestern China, with deficient water resources and a fragile ecology [2–4]. The discharge of mine water may lower the surface water quality and further exacerbate the water shortage in the study area [5,6]. The goaf can store the mine water underground and avoid the intense evaporation loss of water on the ground [7]. The gangue in goaf can adsorb some of the pollutants and remove most of the suspended solids, and thus purify the water [8]. To solve the water shortage problem, thirty-three underground reservoirs have been built in the goaf of the Shendong mining

area in the past two decades. These underground reservoirs have a maximum total storage capacity of $3.1 \times 10^7$ m$^3$ and supply more than 95% of water consumption in the mining area, which well meets the water demand for production, living, and ecology and saves approximately one billion yuan annually [1–4].

Safety, storage capacity, and water quality are three major concerns for an underground reservoir in a coal mine. An underground reservoir should be able to store and purify a large amount of mine water safely and effectively and supply clean water conveniently. Research works related to this have been reported. The goaf, serving as the groundwater reservoir area, has a specific storage coefficient of approximately 0.3, and the caving rocks in goaf have a natural purification effect on the mine water [9,10]. The specific storage coefficient of goaf is primarily dependent on the mining height and rock bulking coefficient and it is the largest in the goaf boundary area [11]. Water affects the deformation and failure characteristics of coal, which should be considered in designing waterproof coal pillar dams [12,13]. Groundwater chemistry varies with temperature and oxygen content and may weather the rock mass and lead to cavern instability and erode the equipment in the underground storage cavern [14]. Exposure to high-sulfate mine water and drying/wetting cycles may lower the durability of artificial concrete dams of underground reservoirs [15]. The oxidation of the sulphides (mainly pyrite) associated with coal generates acid mine drainage [16]. Physico-chemical characteristics of water are key factors to determine its use [17,18] and they can also be used to analyze the sources of mine water [19,20]. The water quality types of Quaternary aquifer and Jurassic burnt rock aquifer in the Shendong mining area are mainly HCO$_3$-Ca type, the mineralization degree of mine water increases, and the water quality types become complicated with the increase in burial depth [21]. The mine water in goaf is widely used to prepare the emulsion of hydraulic props in the Shendong mining area and the high total hardness and sulfate ion concentration of mine water would adversely affect the stability of the emulsion [22]. Owing to the dissolution of fluoride minerals in the Yan'an Formation Jurassic System, the fluorine content of mine water is high in the northwestern Shendong mining area [23]. The high concentration of F$^-$ in mine water is closely related to the dissolution of fluoride-bearing minerals, ion exchange of F$^-$–OH$^-$ in a highly alkaline water environment, and evaporation in the study area [24]. Mine water chemistry varies with space and time [25] and mine water quality can be comprehensively evaluated by some quantitative methods. A Canadian Council of Ministers of the Environment Water Quality Index (CCME WQI) method integrates three evaluation variables into one index and can comprehensively and intuitively show the overall water quality [26,27]. The CCME WQI can flexibly select the corresponding water quality parameter index according to the water use and it has significant advantages in the calculation amount of the evaluation process, the intuitiveness of the evaluation results, and the fault tolerance of the missing values [28,29]. Compared with the Nemerow index and the single factor index, CCME WQI can deliver more stringent results in the presence of a number of over-standard parameters [26].

As a booming technology, the theory and method of underground reservoir still need to be improved. There are few research reports on the distribution characteristics of water quality in the study area and on the significance of water quality for the site selection of an underground reservoir in coal mines. This research focused on the hydrochemistry of mine water and its significance for the site selection of an underground reservoir in the Shendong coal mining area. The hydrochemistry and isotope of mine water samples were measured, and the distribution characteristics of water quality over the mining area were evaluated using a CCME WQI method. Based on the stable isotope composition of hydrogen and oxygen, the source of groundwater in the mining area and the formation mechanism of main excessive ions were analyzed by hydrogeochemical numerical simulation. Combined with the sedimentary characteristics and mining conditions, the influencing factors of the spatial distribution characteristics of water quality in the mining area were analyzed. The significance of water quality for the site selection of underground reservoirs was discussed.

The outcomes may provide references for the site selection of distributed underground reservoirs in similar mining areas with water shortage.

## 2. Materials and Method

### 2.1. Geological Setting

The Shendong coal mining area is located in the border area between the northern Shaanxi Province and the southern Inner Mongolia Autonomous Region. It is located in the southeast of Ordos Plateau, the north edge of Loess Plateau in the northern Shaanxi, and the southeast edge of the Mu Us Desert. The overall landform is high in the northwest and low in the southeast. The altitude is generally 1000~1300 m above sea level with an average altitude of about 1200 m above sea level. The mining area is in a continental monsoon climate zone with an average annual precipitation of 300~400 mm and an average annual evaporation of 2000~2500 mm. There are two types of landforms in the mining area: aeolian sand and loess. The strata exposed in this area are Triassic Yanchang Formation ($T_{3y}$), Jurassic Yan'an Formation ($J_{1-2y}$), Zhiluo Formation ($J_{2z}$), Anding Formation ($J_{2a}$), Cretaceous (K), and Quaternary (Q) (Figure 1). The main surface water systems in the mining area are the Wulanmulun (WL) River and the Boniuchuan River, and the main aquifers are the loose phreatic aquifer of the Quaternary Salawusu Formation, the Cretaceous, and Jurassic sandstone fissure confined aquifers, and the burnt rock confined aquifer in the coal-bearing strata of the Jurassic Yan'an Formation. The lithology of the lower part of the lower Cretaceous Zhidan Formation ($K_{1zh}$) is purple-red and gray-yellow coarse sandstone, conglomerate-bearing sandstone, conglomerate, and so on. The gravel consists of granite, granite gneiss, quartzite, and so on. The rocks have an argillaceous cementation and a loose texture. The lithology of the upper part of the lower Cretaceous Zhidan Formation ($K_{1zh}$) is purple and pink medium- and fine-grained sandstone, siltstone, and sandy mudstone, with an argillaceous cementation and a loose texture. The lower Cretaceous Zhidan Formation ($K_{1zh}$) has an average thickness of 58.40 m. The Yan'an Formation of the Middle-Lower Jurassic System is the main coal-bearing strata and consists of fine sandstone, siltstone, sandy mudstone, a small amount of mudstone, and medium-coarse grained sandstone. The Yan'an Formation generally bears 10~27 coal seams and the minable coal seams from shallow to deep are $1^{-2}$, $2^{-2}$, $3^{-1}$, $4^{-2}$, and $5^{-2}$ coal seams in the mining area. The coal seams usually have a single texture and occasionally contain one or more layers of gangue with a thickness of 0.05~0.62 m. Moreover, the lithology of gangue is mainly peat and sandy mudstone. The minable coal seams belong to medium and thick coal seams. Coal seams are generally shallowly buried in the mining area. The distance between $1^{-2}$ coal seam and $5^{-2}$ coal seam is approximately 170 m. The burial depth of the uppermost coal seam is on average 70 m in the mining area. The burial depth of coal seam is the greatest in the western boundary of the mining area, where $1^{-2}$ coal seam has a burial depth of approximately 150 m. There are many deep valleys in the mining area and each coal seam is exposed by these valleys.

### 2.2. Sampling and Testing

Twenty-six water samples were collected in this study area, including the surface WL River water, coal seam roof water, and inlet water and outlet water of underground reservoirs. Each sample was collected and sealed in a 2.50 L polyethylene bottle. Before sampling, the sample bottle was washed three times with the water sample to be collected. The collected samples were filtered with a 0.45 μm polyethersulfone membrane. Each sample was divided into three subsamples, which were used for cation, anion, and stable isotope analysis, respectively. The samples used for cation (except Hg) analysis were acidified to pH = 2 by adding a few drops of ultrapure $HNO_3$. The samples were sent to the State Key Laboratory of Environmental Geochemistry, Institute of Geochemistry, Chinese Academy of Sciences for a water quality test within 48 h after collection and they are kept at 4 °C until being analyzed. The test items included general chemical indicators (pH, TDS, $HCO_3^-$, $SO_4^{2-}$, $Cl^-$, $F^-$, $NO_3^-$, $Na^+$, $K^+$, $Ca^{2+}$, $Mg^{2+}$, Fe, and Mn), toxicological

indicators (As, Hexavalent Cr, Cd, Pb, and Hg), and stable isotopes of hydrogen and oxygen. The concentrations of conservative anions were measured by ion chromatography and the concentrations of conservative cations were analyzed by flame atomic absorption spectrometry. The pH value was analyzed with a glass electrode method. Toxicological indexes were analyzed by inductively coupled plasma mass spectrometry. The hydrogen and oxygen stable isotopes were measured with a MAT 253 stable isotope mass spectrometer with the Vienna standard average seawater (VSMOW) as a standard sample. The testing procedures follow the Chinese geological and mineral industry standard "Methods for analysis of groundwater quality (DZ/T 0064-2021)". All of the conservative ions of water samples passed the charge balance test and the relative error was less than 5%.

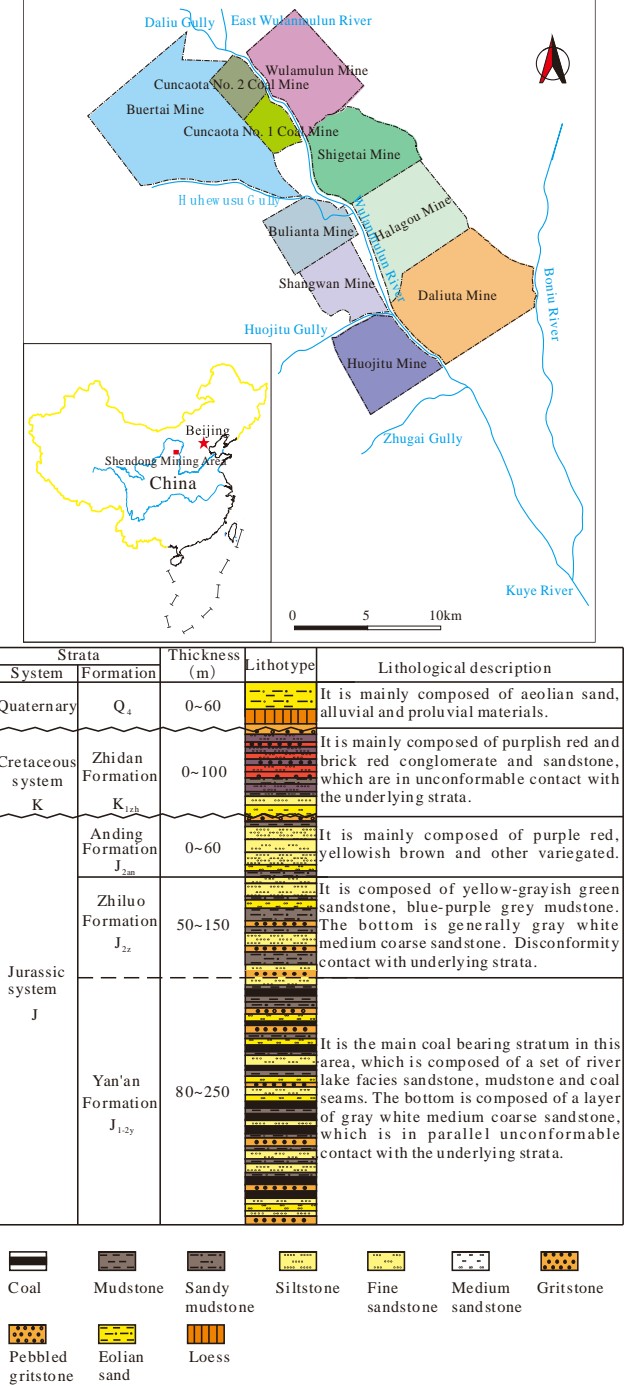

**Figure 1.** Location and stratigraphic histogram of the Shendong mining area.

### 2.3. Evaluation Method of Water Quality

The water quality was quantitatively evaluated using the Canadian Council of Ministers of the Environment Water Quality Index (CCME WQI) method. CCME WQI has three evaluation variables, including the over-standard range of tested variables, the over-standard frequency of tested variables, and the over-standard amplitude of tested variables. The CCME WQI method mainly includes the following steps [26,27].

(1) Determination of the evaluation indexes

Evaluation parameters are generally selected in terms of the water chemical type, function, and major pollution factors. The main source of mine water in the study area is groundwater. The possible influencing factors of water quality include the quality of recharge water, water–rock interaction, human activities, and so on. The mine water is mainly used for reclamation irrigation and underground production. In this study, the evaluation index system of CCME-WQI was constructed according to the use and over-standard indexes.

The sodium adsorption ratio (SAR) is an important indicator of irrigation water quality [30]. SAR can be quantitatively formulated as follows.

$$SAR = \frac{\gamma Na^+}{\sqrt{\frac{\gamma Ca^{2+} + \gamma Mg^{2+}}{2}}} \tag{1}$$

where $\gamma NA^+$, $\gamma Ca^{2+}$, and $\gamma Mg^{2+}$ are the ion concentrations of sodium, calcium, and magnesium, respectively, with a unit of meq/L and the unit of SAR is $(meq/L)^{\frac{1}{2}}$.

If SAR exceeds the standard of 18 $(meq/L)^{\frac{1}{2}}$, sodium in water will replace calcium and magnesium in soil, which will decrease soil permeability and damage the soil. When the SAR value is less than 10 $(meq/L)^{\frac{1}{2}}$, the water quality of irrigation water is excellent; when the SAR value is greater than 18 $(meq/L)^{\frac{1}{2}}$, the water quality is not suitable for irrigation. One of the main uses of mine water in the study area is ecological irrigation. According to the national standard for irrigation water quality (GB5084-2021) and combined with the characteristics of water quality components in the study area, SAR, TDS, $SO_4^{2-}$, $Cl^-$, $F^-$, $Na^+$, $Ca^{2+}$, $Mg^{2+}$, Fe, Mn, As, Pb, and Hg were selected as evaluation indexes.

(2) Calculation of the score of CCME-WQI

The score of CCME-WQI is calculated as follows.

$$CCME - WQI = 100 - \frac{\sqrt{F_1^2 + F_2^2 + F_3^2}}{1.732} \tag{2}$$

where $F_1$ represents the exceeding scope of water quality, $F_2$ represents the exceeding frequency, and $F_3$ represents the exceeding amplitude. These three factors can be calculated as follows.

$$F_1 = \frac{100 \times B}{M} \tag{3}$$

where $M$ is the total number of evaluation indexes and $B$ is the total number of failed variables at least once.

$$F_2 = \frac{100 \times R}{N} \tag{4}$$

where $N$ is the total number of tests and $R$ is the number of failed tests.

$$F_3 = \frac{100 \sum_{i=1}^{M} \sum_{j=1}^{K_i} E_{ij}}{\sum_{i=1}^{M} \sum_{j=1}^{K_i} E_{ij} + \sum_{i=1}^{M} K_i} \tag{5}$$

where $i$ and $j$ are the serial numbers of the variable and sample, respectively; $K_i$ is the total test number of the ith variable; and $E_{ij}$ represents the exceeding amplitude of the ith variable of the jth sample.

For variables for which it is better to be smaller, such as chemical oxygen demand (COD),

$$E_{ij} = 0 \ (X_{ij} \leq C_i) \tag{6}$$

$$E_{ij} = \frac{X_{ij} - C_i}{C_i} \ (X_{ij} > C_i) \tag{7}$$

For variables for which it is better to be larger, such as dissolved oxygen,

$$E_{ij} = 0 \ (X_{ij} \geq C_i) \tag{8}$$

$$E_{ij} = \frac{C_i - X_{ij}}{X_{ij}} \ (X_{ij} < C_i) \tag{9}$$

where $C_i$ is the limit value of the ith variable.

Water quality is divided into five categories (Table 1) [26,27].

**Table 1.** Classification of water quality based on CCME-WQI.

| Category | Range of CCME-WQI | Description |
|---|---|---|
| Excellent | [94~100] | The water quality is in good condition without threat or damage, and the water body has not been damaged. |
| Good | [79~94] | The water quality is in good condition with only a slight threat or damage. The water body is rarely damaged. |
| Fair | [64~79] | The water quality condition is ordinary, the threat or damage is ordinary, and the water body is sometimes polluted and damaged to a certain extent. |
| Marginal | [44~64] | The water quality is poor, the threat or damage is high, and the water body is often polluted and damaged to a large extent. |
| Poor | [0~44] | The water quality condition is very poor, the threat or damage is very high, and the water body is usually polluted and damaged to a great extent. |

## 3. Results and Discussion

### 3.1. Hydrochemical Characteristics of Mine Water

The letters in the serial number of water samples are the initials of the coal mine name, that is, BLT for the Bulianta coal mine, DLT for the Daliuta coal mine, SGT for the Shigetai coal mine, HLG for the Halagou coal mine, SW for the Shangwan coal mine, BET for the Buertai coal mine, and WL for the Wulanmulun coal mine. The SGT, WL, DLT, and HLG mines lie to the east of the WL River and the BET, SW, and BLT mines lie to the west of the WL River (Figure 2). WL-R1, BLT-R1, and DLT-R1 denote the WL river water samples collected from at a site near the WL coal mine, the BLT coal mine, and the DLT coal mine, respectively.

The results of hydrochemical tests (Figure 3 and Table 2) showed that the pH values of water samples in the study area ranged from 7.30 to 9.51, with an average of 7.90; the $HCO_3^-$ concentration was 166.50~1304.10 mg/L, with an average of 508.82 mg/L; the $Cl^-$ concentration was 4.67~980.29 mg/L, with an average of 140.82 mg/L; the $SO_4^{2-}$ concentration was 12.73~951.71 mg/L, with an average of 234.36 mg/L; the $F^-$ concentration was 0.23~11.65 mg/L, with an average of 2.29 mg/L; the $Na^+$ concentration was 10.53~1292.79 mg/L, with an average of 354.11 mg/L; the $Ca^{2+}$ concentration was 1.69~146.33 mg/L, with an average of 54.56 mg/L; the TDS concentration was 220.99~3370.47 mg/L, with an average of 1023.57 mg/L; and the SAR was 0.316~91.43 $(\text{meq/L})^{\frac{1}{2}}$, with an average of 21.35 $(\text{meq/L})^{\frac{1}{2}}$. Overall, the concentrations of groundwater conservative ions in the mine area follow a descending order: $HCO_3^- > SO_4^{2-} > Cl^-$, $Na^+ + K^+ > Ca^{2+} > Mg^{2+}$ (Figure 3).

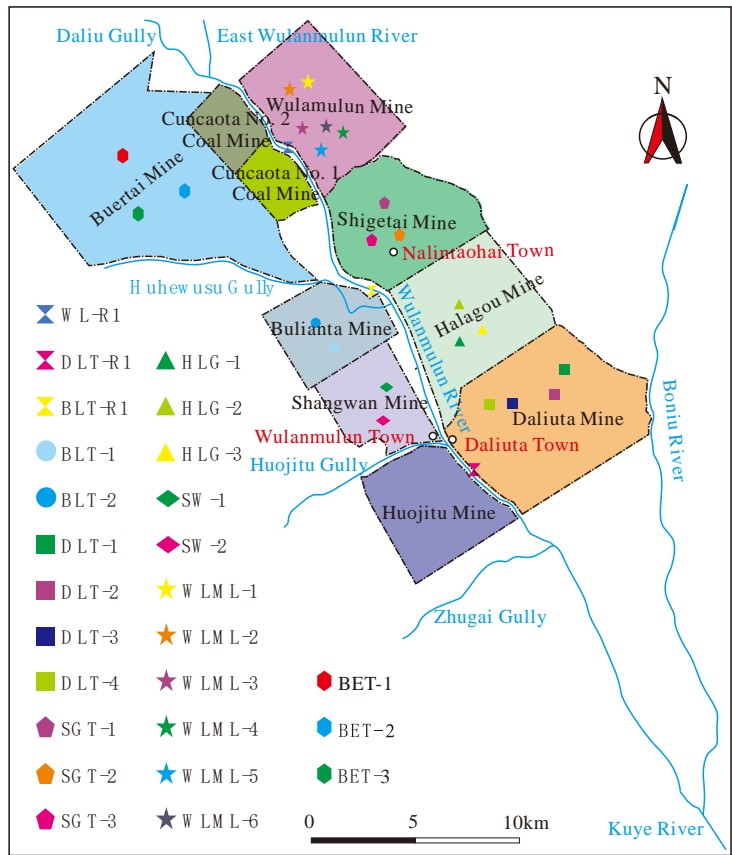

**Figure 2.** Sampling points in the study area.

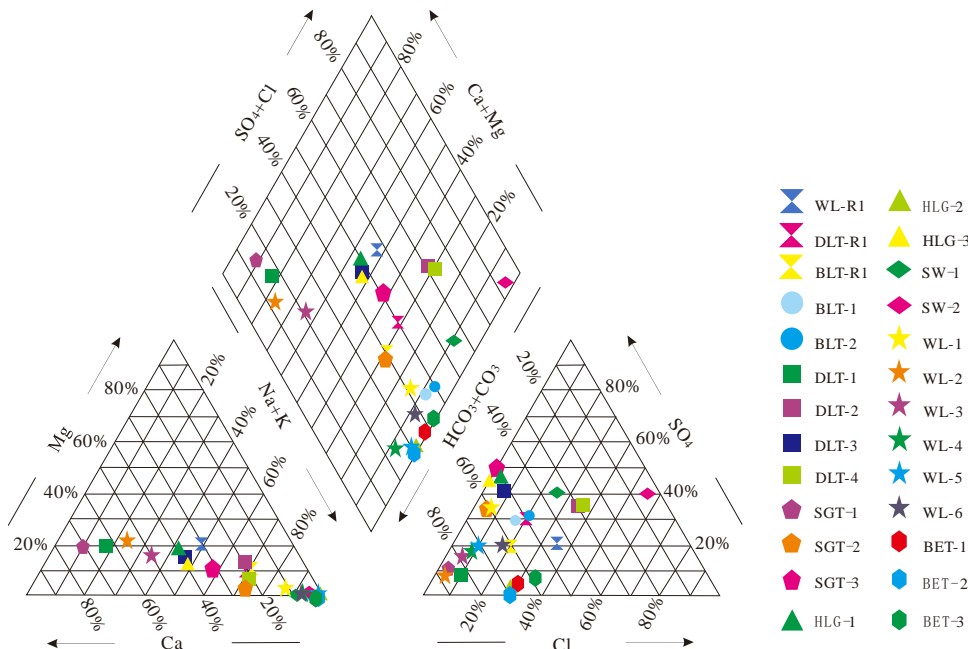

**Figure 3.** Piper graph of the water quality test results for the water samples of the study area.

**Table 2.** Statistics of water quality test results.

| Variable | Test Results (mg/L) | | | National Standard (mg/L) | | | | Over-Limit Conditions | |
|---|---|---|---|---|---|---|---|---|---|
| | Min | Max | Average | Surface Water III | Drinking Water | Groundwater Class III | Irrigation Water | Number of Over-Limit Tests | Over-Limit Ratio (%) |
| pH | 7.30 | 9.51 | 7.90 | 6.0~9.0 | 6.5~8.5 | 6.5~8.5 | 5.5~8.5 | 1 | 3.85 |
| TDS | 220.99 | 3370.47 | 1023.57 | 1000 | 1000 | 1000 | 1000 (non-saline-alkali soil area) 2000 (saline-alkali soil area) | 10 | 38.46 |
| $HCO_3^-$ | 166.50 | 1304.1 | 508.82 | — | — | — | — | — | — |
| $SO_4^{2-}$ | 12.73 | 951.71 | 234.36 | 250 | — | 250 | — | 11 | 42.31 |
| $Cl^-$ | 4.67 | 980.29 | 140.82 | 250 | — | 250 | 350 | 5 | 19.23 |
| $F^-$ | 0.23 | 11.65 | 2.29 | 1 | — | — | 2 (general area) 3 (high fluorine area) | 15 | 57.69 |
| $Na^+$ | 10.53 | 1292.79 | 354.11 | — | 200 | 200 | — | 17 | 65.38 |
| $K^+$ | 1.01 | 8.68 | 4.36 | — | — | — | — | — | — |
| $Ca^{2+}$ | 1.69 | 146.33 | 54.56 | — | — | — | — | — | — |
| $Mg^{2+}$ | 0.03 | 40.05 | 12.92 | — | — | — | — | — | — |
| Fe | 0.0020 | 17.31 | 0.862 | — | 0.3 | 0.3 | — | 4 | 15.40 |
| Mn | — | 3.16 | 0.29 | — | 0.1 | 0.1 | — | 11 | 42.31 |
| $NO_3^-$ | — | 5.1613 | 0.64 | 10 | — | 20 | — | 0 | 0 |
| As | $7.95 \times 10^{-5}$ | $4.44 \times 10^{-2}$ | $5.69 \times 10^{-3}$ | 0.05 | 0.01 | 0.01 | 0.1 | 2 | 15.40 |
| Cr | $1.64 \times 10^{-4}$ | $1.55 \times 10^{-3}$ | $4.39 \times 10^{-4}$ | 0.05 | 0.05 | 0.05 | 0.1 | 0 | 0 |
| Cd | $1.32 \times 10^{-6}$ | $7.03 \times 10^{-4}$ | $1.36 \times 10^{-4}$ | 0.005 | 0.005 | 0.005 | 0.01 | 0 | 0 |
| Pb | $5.76 \times 10^{-7}$ | $3.15 \times 10^{-2}$ | $2.02 \times 10^{-3}$ | 0.05 | 0.01 | 0.01 | 0.2 | 1 | 3.85 |
| Hg | $2.85 \times 10^{-6}$ | $1.66 \times 10^{-4}$ | $2.61 \times 10^{-5}$ | 0.0001 | 0.001 | 0.001 | 0.001 | 1 | 3.85 |
| SAR | 0.32 | 91.43 | 21.35 | — | — | — | — | 8 | 42.31 |

According to the Shukarev's classification, there are ten water chemical types in the study area. Among the twenty-six water samples, five samples belong to the HCO$_3$·SO$_4$–Na type, five to the HCO$_3$–Na type, four to the HCO$_3$·SO$_4$–Na·Ca type, three to the HCO$_3$–Ca type, three to the HCO$_3$·Cl–Na type, two to the SO$_4$·Cl·HCO$_3$–Na type, one to the SO$_4$·HCO$_3$·Cl–Na type, one to the HCO$_3$·Cl–Na·Ca type, one to the HCO$_3$–Ca·Na type, and one to the Cl·SO$_4$–Na type.

A Chadha diagram can be used to determine the hydrochemical type of groundwater and the results are primarily consistent with the analysis results of the Piper diagram and the Shukarev's classification [31]. The Chadha diagram shows that the groundwater in the Shendong mining area mainly consists of NaHCO$_3$ (56.5%, reverse ion exchange), CaHCO$_3$ (21.7%, weathering and recharge), and NaCl (17.4%, evaporation) types (Figure 4).

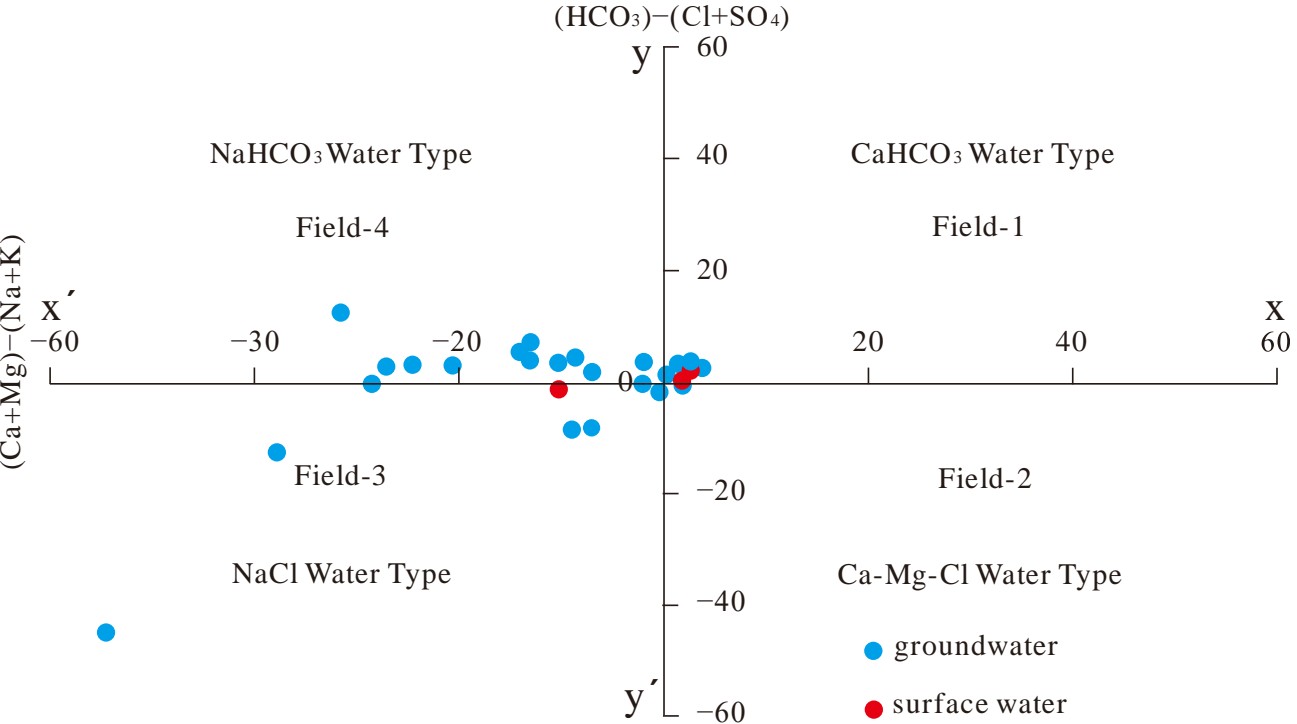

**Figure 4.** Chadha diagram of the classification of groundwater types in the Shendong mining area.

According to China National Environmental quality standards for surface water (GB3838-2002) (Class III water), standards for drinking water quality (GB5749-2006), standards for groundwater quality (GB/T 14848-2017) (Class III water), and standards for irrigation water quality (GB5084-2021), the over-limit variables include pH, TDS, SO$_4^{2-}$, Cl$^-$, F$^-$, Na$^+$, Fe, Mn, As, Pb, Hg, and SAR. Some failed variables have a high exceeding frequency factor, including Na$^+$ (the exceeding frequency factor F$_2$ is 65.38), F$^-$ (57.69), SAR (42.31), SO$_4^{2-}$ (42.31), Mn (42.31), and TDS (38.46) (Table 2).

The correlation analysis was performed for the tested variables (Table 3) and the results showed that the following variables had strong positive correlations with a correlation coefficient greater than 0.85: TDS versus Cl$^-$, TDS versus SO$_4^{2-}$, TDS versus Na$^+$, Cl$^-$ versus Na$^+$, SO$_4^{2-}$ versus Na$^+$, Ca$^{2+}$ versus Mg$^{2+}$, and F$^-$ versus SAR. A certain negative correlation with a correlation coefficient of −0.653 exists between F$^-$ and Ca$^{2+}$, which is mainly due to the fact that the two elements can chemically combine to form water-insoluble CaF$_2$ precipitates (Pan et al., 2021; Zhang et al., 2021); therefore, water samples with a high F$^-$ concentration have a low Ca$^{2+}$ concentration.

**Table 3.** Pearson correlation coefficients of water quality indexes for water samples in the study area.

| | pH | TDS | $HCO_3^-$ | $SO_4^{2-}$ | $Cl^-$ | $F^-$ | $Na^+$ | $K^+$ | $Ca^{2+}$ | $Mg^{2+}$ | SAR |
|---|---|---|---|---|---|---|---|---|---|---|---|
| pH | 1.000 | | | | | | | | | | |
| TDS | 0.242 | 1.000 | | | | | | | | | |
| $HCO_3^-$ | 0.075 | 0.376 | 1.000 | | | | | | | | |
| $SO_4^{2-}$ | 0.232 | 0.968 ** | 0.303 | 1.000 | | | | | | | |
| $Cl^-$ | 0.214 | 0.877 ** | −0.066 | 0.824 ** | 1.000 | | | | | | |
| $F^-$ | 0.355 | 0.543 ** | 0.340 | 0.355 | 0.549 ** | 1.000 | | | | | |
| $Na^+$ | 0.272 | 0.975 ** | 0.399 | 0.900 ** | 0.872 ** | 0.670 ** | 1.000 | | | | |
| $K^+$ | −0.129 | 0.405 | 0.430 * | 0.459 * | 0.167 | −0.059 | 0.291 | 1.000 | | | |
| $Ca^{2+}$ | −0.179 | −0.047 | −0.129 | 0.150 | −0.141 | −0.653 ** | −0.263 | 0.408 | 1.000 | | |
| $Mg^{2+}$ | −0.203 | −0.155 | −0.182 | −0.013 | −0.167 | −0.578 ** | −0.349 | 0.454 * | 0.868 ** | 1.000 | |
| SAR | 0.422 | 0.703 ** | 0.348 | 0.551 * | 0.676 ** | 0.932 ** | 0.818 ** | −0.035 | −0.619 | −0.627 | 1.000 |

Note: ** and * indicate the significance levels of $p < 0.01$ and $p < 0.05$, respectively.

The CCME-WQI scores of the samples in the study area range from 37.46 to 100.00, with an average of 72.76. The average CCME-WQI score of the mine water follows a descending order as follows: SGT (average CCME-WQI = 88.32) > WL (86.49) > DLT (75.63) > HLG (67.16) > BET (57.71) > SW (41.57) > BLT (37.65) (Figure 2, Table 4). The CCME-WQI scores of mine water are relatively low at the BET (57.71), SW (41.57), and BLT (37.65) mines and there are a total of 11 over-limit variables at these mines, including pH, SAR, TDS, Mn, $Na^+$, $F^-$, Cl, $SO_4^{2-}$, Hg, Pb, and As. On the plane, the mine water quality is better on the east of the WL River than on the west of the WL River.

The mine water samples from the roof of the shallow coal seams, e.g., $1^{-2}$ and $2^{-2}$ coal seams of the WLML, SGT, HLG, and DLT mines on the east of the WL River, have a CCME-WQI score of 100.00, indicating that the water quality of the shallow aquifers in this region is good. The CCME-WQI scores of the mine water from the $3^{-1}$ coal seam roof are significantly lower than that from the $1^{-2}$ and $2^{-2}$ coal roofs at the SGT and WLML mines (Table 4), indicating that the mine water quality tends to decrease with the increase in mining depth on the east of the WL River.

The WL River runs roughly from north to south within the study area and the CCME-WQI score of surface river water samples at the WL, BLT, and DLT mines are 100.00, 86.15, and 86.21, respectively. The CCME-WQI score gradually declines from north to south. That is, the water quality of the WL River tends to decrease as it flows through the Shendong mining area, which is due to the inflow of some other over-limit waters.

Field 1: $CaHCO_3$ water type representing weathering and recharge. Field 2: Ca-Mg-Cl water type shows reverse ion exchange. Field 3: NaCl water type representing evaporation processes. Field 4: $NaHCO_3$ water type shows ion exchange processes.

### 3.2. Origins of Sodium and Fluoride Ions in the Waters in the Study Area

Sodium ions and fluorine ions have a high exceeding frequency in the mining area. It is of great significance to investigate the genetic mechanism of these two ions for the scientific treatment and utilization of mine water. The correlation between $\gamma Na^+ + \gamma K^+$ and $\gamma Cl^-$ can reflect the ion source [32,33], where $\gamma$ represents the ion concentration with a unit of meq/L. When the ratio of the two variables, $(\gamma Na^+ + \gamma K^+)/\gamma Cl^-$, is equal to 1, it is generally assumed that $Na^+$ and $K^+$ in groundwater mainly come from the dissolution of rock salt and potassium salt. When this ratio is greater than 1, it indicates that the contents of $Na^+$ and $K^+$ in groundwater are affected by the dissolution of silicate minerals. The relationship between $\gamma Na^+ + \gamma K^+ - \gamma Cl^-$ and $\gamma Ca^{2+} + \gamma Mg^{2+} - \gamma SO_4^{2-} - \gamma HCO_3^-$ can be used to indicate the cation exchange effect. When the proportional coefficient of these two variables is equal to −1, it is generally assumed that the cation exchange effect has a significant influence on water quality [20,34].

**Table 4.** CCME-WQI scores and over-limit indexes of the samples.

| Mine Field | Average CCME-WQI | Sample Number | Coal Seam Related | Sampling Point | Over-Limit Indexes | CCME-WQI | Grade |
|---|---|---|---|---|---|---|---|
| WL River | 90.79 | WL-R1 | - | WL riverside at WL mine | − | 100 | Excellent |
| | | BLT-R1 | - | WL riverside at BLT mine | $F^-$, $Na^+$ | 86.15 | Good |
| | | DLT-R1 | - | WL riverside at DLT mine | $F^-$, $Na^+$ | 86.21 | Good |
| BET | 57.71 | BET-1 | $1^{-2}$ | Roof | pH, SAR, $F^-$, $Na^+$ | 65.29 | Fair |
| | | BET-2 | $4^{-2}$ | Roof | SAR, TDS, $Na^+$, $F^-$, $Cl^-$, Fe, Mn | 58.64 | Marginal |
| | | BET-3 | $2^{-2}$ | Roof | SAR, TDS, $Na^+$, $F^-$, $Cl^-$, Fe, Mn, As | 49.21 | Marginal |
| SW | 41.57 | SW-1 | $2^{-2}$ | Goaf | SAR, TDS, Mn, $Na^+$, $F^-$, $Cl^-$, $SO_4^{2-}$ | 42.42 | Poor |
| | | SW-2 | $2^{-2}$ | Roof | SAR, TDS, $Na^+$, $F^-$, $Cl^-$, $SO_4^{2-}$, Hg | 40.72 | Poor |
| BLT | 37.65 | BLT-1 | $2^{-2}$ | Goaf | SAR, TDS, Fe, Mn, $Na^+$, $F^-$, $SO_4^{2-}$, Pb | 37.84 | Poor |
| | | BLT-2 | $2^{-2}$ | Inlet of underground reservoir | SAR, TDS, Fe, Mn, $Na^+$, $F^-$, $SO_4^{2-}$, As | 37.46 | Poor |
| HLG | 67.16 | HLG-1 | $2^{-2}$ | Goaf effluent | $SO_4^{2-}$, Mn | 57.49 | Marginal |
| | | HLG-2 | $2^{-2}$ | Underground reservoir | $SO_4^{2-}$, Mn | 71.22 | Fair |
| | | HLG-3 | $3^{-1}$ | Roof | SAR, $F^-$, $Na^+$ | 72.77 | Fair |
| DLT | 75.63 | DLT-1 | $2^{-2}$ | Roof | − | 100 | Excellent |
| | | DLT-2 | $2^{-2}$ | Goaf | TDS, $SO_4^{2-}$, $F^-$, $Na^+$, Mn | 69.62 | Fair |
| | | DLT-3 | $2^{-2}$ | Goaf | TDS, $SO_4^{2-}$, $Na^+$, Mn | 61.69 | Marginal |
| | | DLT-4 | $2^{-2}$ | Underground reservoir | TDS, $SO_4^{2-}$, $F^-$, $Na^+$, Mn | 71.19 | Fair |
| SGT | 88.32 | SGT-1 | $2^{-2}$ | Roof | − | 100 | Excellent |
| | | SGT-2 | $3^{-1}$ | Goaf | TDS, $SO_4^{2-}$, $Na^+$, Mn | 71.8 | Fair |
| | | SGT-3 | $3^{-1}$ | Inlet of underground reservoir | $SO_4^{2-}$ | 93.17 | Good |
| WL | 86.49 | WL-1 | $1^{-2}$ | Goaf | − | 100 | Excellent |
| | | WL-2 | $1^{-2}$ | Roof | − | 100 | Excellent |
| | | WL-3 | $3^{-1}$ | Goaf | − | 100 | Excellent |
| | | WL-4 | $3^{-1}$ | Underground reservoir | SAR, $F^-$, $Na^+$ | 71.36 | Fair |
| | | WL-5 | $3^{-1}$ | Roof | SAR, $F^-$, $Na^+$ | 76.21 | Fair |
| | | WL-6 | $3^{-1}$ | Inlet of underground reservoir | SAR, $F^-$, $Na^+$, As | 71.38 | Fair |

For most of the mine water samples in the study area, the ratio of $\gamma Na^+ + \gamma K^+$ to $\gamma Cl^-$ is much larger than 1 (Figure 5a), which indicates that $Na^+$ is not only related to the dissolution of rock salt, but also related to the dissolution of silicate minerals in the strata of the study area. Albite dissolves to produce $Na^+$ and its chemical reaction formula is as follows:

$$2NaAlSi_3O_8 + 2CO_2 + 11H_2O = Al_2Si_2O_5[OH]_4 + 2Na^+ + 4H_4SiO_4 + 2HCO_3^- \tag{10}$$

For the water samples in the study area, a good negative correlation exists between $(\gamma Na^+ + \gamma K^+ - \gamma Cl^-)$ and $(\gamma Ca^{2+} + \gamma Mg^{2+} - \gamma SO_4^{2-} - \gamma HCO_3^-)$. The ratio coefficient of $(\gamma Na^+ + \gamma K^+ - \gamma Cl^-)$ to $(\gamma Ca^{2+} + \gamma Mg^{2+} - \gamma SO_4^{2-} - \gamma HCO_3^-)$ is $-1.16$ and the goodness of fit $R^2 = 0.98$ (Figure 5b), which indicates that the cation exchange has a significant impact on the chemical composition of water samples. The Gibbs diagram also shows that the mine water quality in the study area is mainly controlled by the weathering and dissolution of rocks, and the evaporation and concentration effect is relatively weak (Figure 6) [35].

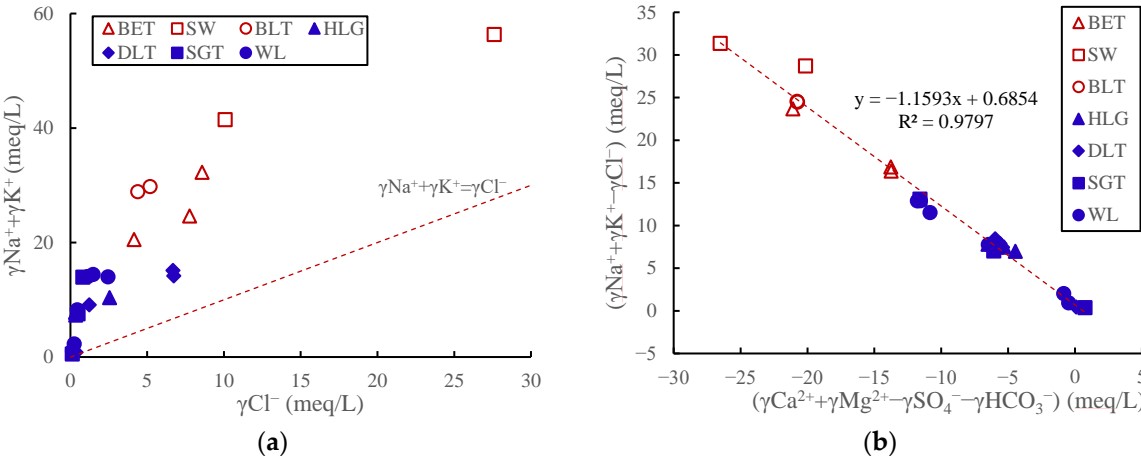

**Figure 5.** Ion's ratio of mine water samples. (**a**) ($\gamma Na^+ + \gamma K^+$) vs. $\gamma Cl^-$. (**b**) ($\gamma Na^+ + \gamma K^+ - \gamma Cl^-$) vs. ($\gamma Ca^{2+} + \gamma Mg^{2+} - \gamma SO_4^{2-} - \gamma HCO_3^-$).

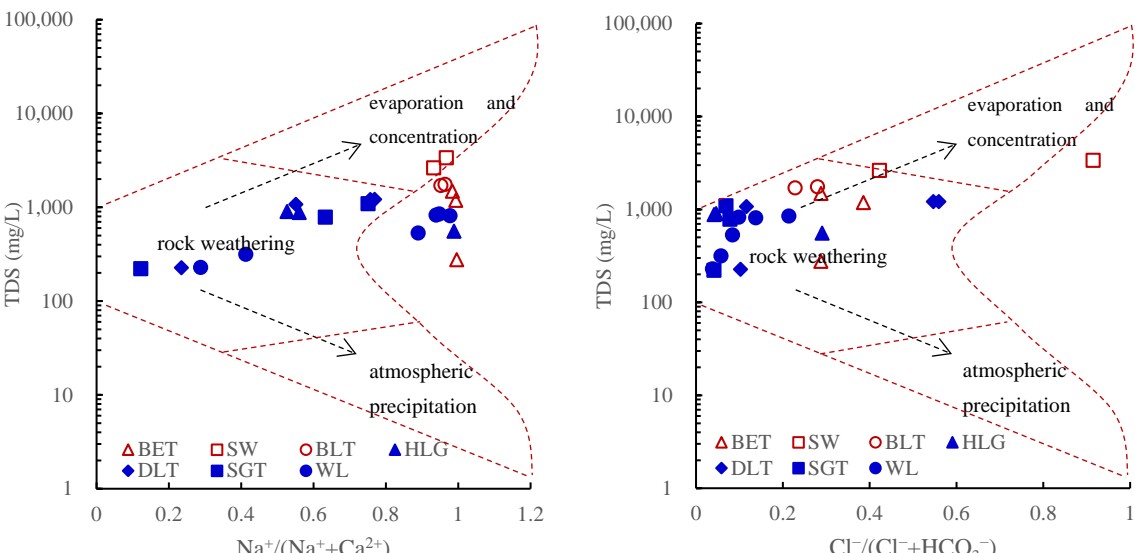

**Figure 6.** Gibbs diagram of mine water samples in the study area.

A scatter diagram of $\gamma Na^+ + \gamma K^+$ versus $\gamma Cl^-$ can imply the direction of cation exchange. When the data point is below the 1:1 line, it indicates that $Na^+$ in water decreases and $Ca^{2+}$ increases, resulting in positive cation exchange. When the data point is above the 1:1 line, it shows that $Na^+$ increases and $Ca^{2+}$ decreases in water, and reverse cation exchange occurs [20,32–34]. The mine water in the study area falls above the 1:1 line (Figure 5a), indicating that reverse cation exchange occurs. That is, the exchange between $K^+$ and $Na^+$ on the surface of rock/soil and $Ca^{2+}$ and $Mg^{2+}$ in water leads to the increase in $Na^+$ concentration and the decrease in $Ca^{2+}$ concentration in water, which promotes the dissolution of fluorine-containing minerals, such as fluorite ($CaF_2$) and fluorapatite [$Ca_5(PO_4)_3F$], and thus increases the $F^-$ concentration in the mine water.

Fluorine is an active element and its natural occurrence is rather complicated. Fluorine can be adsorbed on the surface of clay, zeolite, mica, and other minerals. Fluorine also exists in the form of compounds, including fluorite, apatite, diorite, and other minerals [34,36]. The cation exchange effect has a significant influence on the hydrochemical compositions of water. The exchange adsorption capacity of hydrogen ion is higher than that of the other monovalent, divalent, and trivalent cations. A high concentration of $H^+$ in water is not conducive to the exchange adsorption of other cations between water and solid minerals.

The higher the pH value, the stronger the cation exchange performance between rock/soil and water. When the pH value increases from 6 to 11, the exchange capacity increases by 1~2 times. In an alkaline environment, the concentration of $OH^-$ in water is high and $Ca^{2+}$ in water can easily react with $OH^-$ to form $Ca(OH)_2$ precipitation, which leads to the decrease in the $Ca^{2+}$ concentration in water and promotes the dissolution of fluorite or fluorapatite, and leads to the increase in the $F^-$ concentration in water. In addition, when the pH value is less than the zero potential $pH_z$ value (the surface charge is zero), the surface of the rock and soil particles is positively charged, and has a certain adsorption for $F^-$, $PO_4^{3-}$, and other anions. In an alkaline environment, $OH^-$ in water replaces $F^-$ adsorbed on the surface of the rock and soil particles, so that the concentration of $F^-$ in water increases. For example, $F^-$ in apatite, muscovite, and biotite exchanges with $OH^-$ in water.

Fluoridated hydroxyapatite:

$$Ca_5(PO_4)_3F + OH^- \rightarrow Ca_5(PO_4)_3OH + F^- \tag{11}$$

Muscovite:

$$KAl_2[AlSi_3O_{10}]F_2 + 2OH^- \rightarrow KAl_2[AlSi_3O_{10}][OH]_2 + 2F^- \tag{12}$$

Biotite:

$$KMg_3[AlSi_3O_{10}]F_2 + 2OH^- \rightarrow KMg_3[AlSi_3O_{10}][OH]_2 + 2F^- \tag{13}$$

The saturation index (*SI*) is one of the parameters that reflect the water–rock interaction and determine the reaction state between minerals and aqueous solution. The mathematical expression of *SI* is as follows:

$$SI = \log \frac{IAP}{K} \tag{14}$$

where *IAP* is the ionic activity product, dimensionless; *K* is the solubility product constant of a mineral at a specific temperature, dimensionless. When *SI* < 0, the mineral is in an unsaturated state in groundwater and tends to dissolve; when *SI* = 0, mineral and groundwater are in equilibrium; when *SI* > 0, a mineral tends to precipitate [37].

A hydrogeochemical numerical simulation software PHREEQC can simulate the dissolution process of various minerals in water [36]. The software version used in this paper is PHREEQC V3.20 and the database are 'llnl.dat' of WATEQ. The measured temperature, pH, ion concentrations, and other parameters were input into this code. The *SI* of calcite, dolomite, gypsum, halite, and fluorite in mine water samples was obtained by numerical simulation. The range and average of *SI* of calcite, dolomite, fluorite, halite, and gypsum are (−0.40~1.40 and 0.35), (−2.50~2.40 and 0.12), (−2.20~−0.30 and −1.19), (−8.90~−4.60 and −6.49), and (−3.80~−1.00 and −1.99), respectively. It is concluded that fluorite, rock salt, and gypsum minerals in all mine water samples are unsaturated (Figure 7). Dolomite minerals in 43.5% of the samples and calcite in 21.7% of the samples are in an unsaturated state. Calcite in 78.0% of the samples and dolomite in 56.5% of the samples are in a supersaturated state. This indicates that fluorite, halite, gypsum, and other minerals in the rock will be dissolved into the mine water, and the dissolved products will further promote the precipitation of calcite and dolomite. From Figure 7, it can be seen that the concentration of $F^-$ is positively correlated to the dissolution of fluorite and halite. The chemical equations for mineral dissolution and precipitation are as follows:

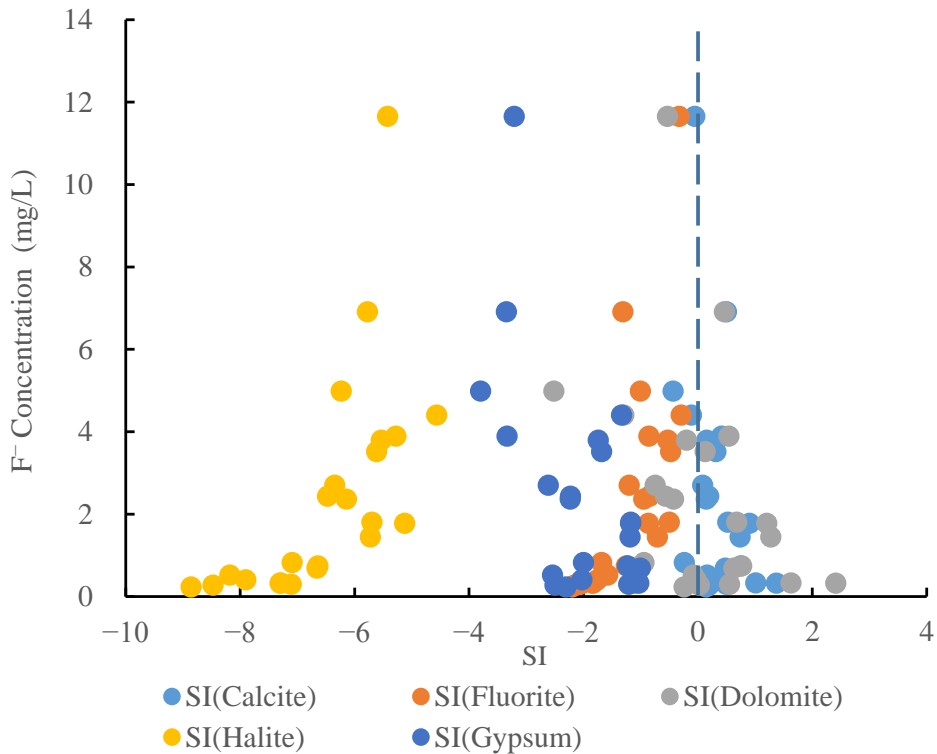

**Figure 7.** Relationship between fluoride ion concentration and mineral saturation index (*SI*).

*3.3. Recharge of Groundwater in the Study Area*

Groundwater is the primary water-filling source of mine pits and the recharge source of groundwater is an important factor affecting the mine water quality in the study area. Hydrogen and oxygen stable isotopes have a good chemical stability and can be used to analyze the recharge sources of groundwater. Liu counted 139 sets of isotopic data of atmospheric precipitation in the northern Ordos Basin and adjacent areas and obtained an equation of the atmospheric precipitation line in the northern Ordos Basin (local meteoric water line, LMWL) as follows: $\delta D = 6.769\delta^{18}O - 0.428$; LMWL has a smaller slope than the global atmospheric precipitation line (GMWL) [32]. The average $\delta D$ and $\delta^{18}O$ of local atmospheric precipitation is $-8.31‰$ and $-56.70‰$, respectively [32]. On the $\delta D$–$\delta^{18}O$ scatter diagram, all of the sample data points are on the right side of the LMWL (Figure 8), reflecting that the samples originate from atmospheric precipitation and show certain characteristics of oxygen drift (relatively rich in $^{18}O$). $\delta D$ ranges from $-85.03‰$ to $-53.93‰$ and $\delta^{18}O$ from $-11.27‰$ to $-6.47‰$. The river water sample is on the upper right of the atmospheric precipitation sample, indicating that the river water experienced a certain evaporation and the heavy isotope is relatively enriched. The mine water samples are located on the lower left of the atmospheric precipitation sample, indicating that the abundance of heavy isotope is lower in mine water (particularly the samples from the BET and SW mines on the west of the WL River) than in modern atmospheric precipitation and the isotopic characteristics of the mine water samples are similar to those of paleo-atmospheric precipitation under a cold and humid paleoclimate [34]. That is, the mine water in the mining area mainly originates from paleo-atmospheric precipitation under a relatively cold and humid paleoclimate and has stayed in the strata for a long time, which is beneficial to mineral dissolution and cation exchange adsorption, particularly the samples from the BET and BLT mines (Figure 8).

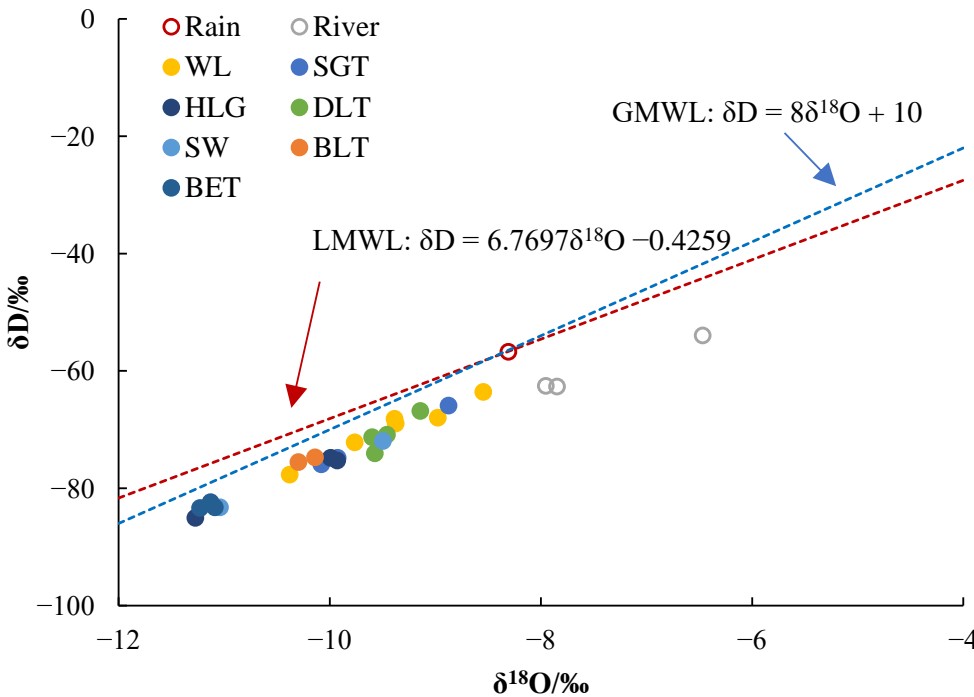

**Figure 8.** Hydrogen and oxygen stable isotopes of water samples in the study area.

*3.4. The Relationship between Stratigraphic Texture and Water Quality*

The middle and lower Jurassic coal-bearing strata in the Shendong mining area belong to lacustrine delta and fluvial sedimentary systems, and the lithology is clastic rock [38]. The Lower Cretaceous Zhidan Group is an important aquifer in the Ordos Basin, but it is unevenly distributed in space [39]. In the study area, the WL River is roughly the dividing line, the Zhidan Group is missing, and the burial depth of coal seam is relatively small on the east of the WL River; the Zhidan group is well-developed and the buried depth of coal seam is relatively large on the west of the WL River. Compared with Jurassic coal-bearing strata, the red strata of the Zhidan Group are rich in Fe, Mn, and other elements. Therefore, the water–rock interaction type and products are also different on the east and on the west of the river.

The possible sources of mine water in the study area include atmospheric precipitation, surface water, Quaternary loose layer water, Cretaceous bedrock fissure water, and Jurassic bedrock fissure water. The possible water-filling channels include unsealed/poorly sealed boreholes, mining-induced fractures, natural bedrock fractures, skylight windows of the water-resisting layer, and burned rock holes.

On the west of the WL River, the lower Cretaceous strata are well-developed and the burial depth of coal seam is relatively large, the thickness of the Zhiluo Formation is large with an average of 70.71 m, and the lithology is mainly sandy mudstone with good water-resisting ability. The water-conducting fracture zone induced by coal mining in Yan'an Formation does not extend to the top of the Zhiluo Formation [4,19]. The water-filling source is generally static reserves of the Jurassic bedrock fissure water. The hydraulic connection between surface water, shallow groundwater, and deep groundwater is relatively weak (Figures 9a and 10a).

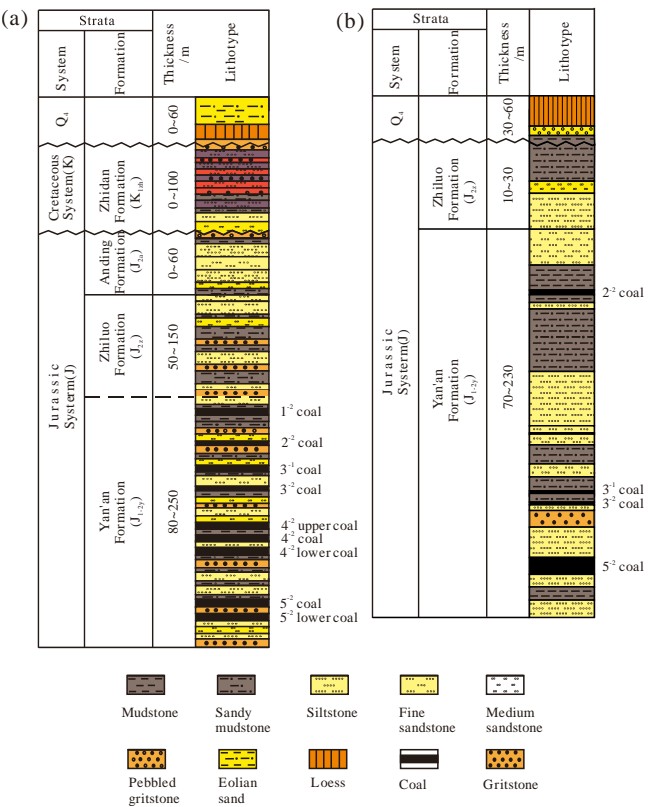

**Figure 9.** Typical stratigraphic texture: (**a**) with the Anding Formation (J$_{2a}$) and Zhidan Formation (K$_{1zh}$) on the west of the WL River and (**b**) without J$_{2a}$ and K$_{1zh}$ on the east of the WL River.

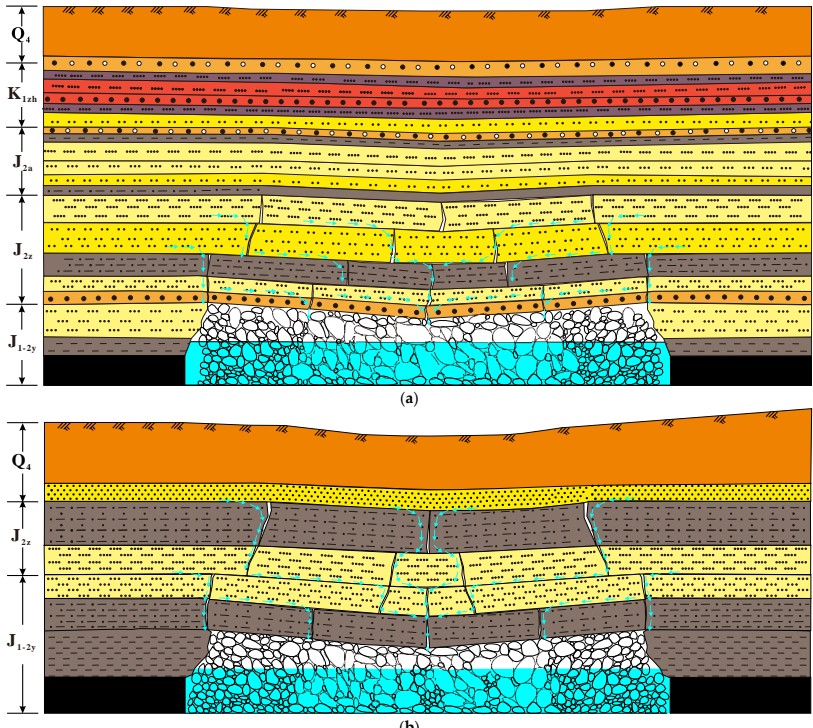

**Figure 10.** Schematic diagram of the height of the water-conducting fracture zone in the study area. (**a**) on the west of the WL River. (**b**) on the east of the WL River.

On the east of the WL River, such as in the DLT, HLG, and other minefields, the thickness of Quaternary is large, with an average thickness of 30–60 m, and the strata lack

the Zhidan Group of the Lower Cretaceous System. The Zhiluo Formation of the Middle Jurassic System is thin, with a thickness of only 10–30 m. The minable coal seams in the Jurassic Yan'an Formation have a relatively small burial depth and a thin bedrock covering. In the process of coal mining, sudden inrushes of water and sand from Quaternary loose layers into the mine pit are prone to occur and the mine water is closely related to shallow Quaternary groundwater (Figures 9b and 10b). Owing to the relatively low mechanical strength of the coal-bearing strata and small burial depth at the DLT mine on the east of WL River, there are only a caving zone and fracture zone above the goaf and the fracture zone extends directly to the surface [2]. The hydraulic connection between surface water, shallow groundwater, and deep groundwater is very close.

In summary, the stratigraphic texture on the east of the WL River is different from that on the west of the WL River. The differences in stratigraphic texture are usually related to tectonic evolution and sedimentary system. Both the stratigraphic texture and mining activity determine the channels and sources of mine water. The water sources and water–rock interaction determine the hydrochemical characteristics of mine water.

*3.5. The Implications of Mine Water Hydrochemistry for the Site Selection of Underground Reservoirs*

Underground reservoirs can mitigate water shortage and relieve environmental pollution. Underground reservoirs can store a huge volume of mine water and avoid the loss from evaporation in the case of surface drainage of mine water. Underground reservoirs can purify the mine water in this way; that is, they can effectively lower the suspended solids, turbidity, electrical conductivity (EC), and total dissolved solids (TDSs) and decrease the concentrations of some organic matters, heavy metals, chemical oxygen demand (COD), $Fe^{3+}$, and $Mn^{2+}$ [9,40], and thus reduce the construction and operation cost of a mine water treatment plant.

Whether an underground reservoir can supply good-quality water depends on the inlet water quality as well as the water–rock interaction. The water–rock interaction in goaf has a possibility of lowering the water quality to some extent in some mine fields. Hydrochemical test results showed that $Na^+$ (the exceeding frequency factor $F_2$ is 65.38), $F^-$ ($F_2$ = 57.69), and SAR ($F_2$ = 42.31) are major failed variables of the mine water samples in the study area (Table 2). The water–rock interactions in goaf are dominantly the dissolution of silicate minerals (such as albite), rock salt, fluorine-containing minerals {such as fluorite ($CaF_2$) and fluorapatite [$Ca_5(PO_4)_3F$]}, and reverse cation exchange adsorption. The results of hydrogeochemical numerical simulation show that fluorite, rock salt, and gypsum minerals are unsaturated in all mine water samples and calcite and dolomite are unsaturated in some water samples (Figure 7). $F^-$ in apatite, muscovite, and biotite can exchange with $OH^-$ in alkaline water, which increases the $F^-$ concentration in water. That is, the water–rock interactions in underground reservoirs increase the $Na^+$, $F^-$, and SAR of mine water, and thus lower the water quality, which must be considered in the site selection of underground reservoirs in the study area. The mine water is primarily used for irrigation in the mining area. Some areas (such as the BET, SW, and BLT mine fields) are not suitable for the construction of an underground reservoir. Because the $Na^+$ and $F^-$ concentrations and SAR in mine water are failed variables in these areas and the storage of mine water in an underground reservoir would further lower the water quality by increasing the values of $Na^+$ and $F^-$ concentrations and SAR and then harm the plants. The areas where the mine water has a low concentration of $Na^+$ and $F^-$ and a small SAR value (such as the shallow coal seams at the SGT, DLT, and WL mines) can be considered as possibly favorable sites for underground reservoirs. $F^-$ and $Ca^{2+}$ can chemically combine to form water-insoluble $CaF_2$ precipitates and lower the $F^-$ concentration in water. To ensure that the underground reservoir can effectively play its role in purifying mine water, the areas where the water has a high *SI* of calcite and dolomite, a low *SI* of fluorite ($CaF_2$) and fluorapatite, and a low $Na^+$ concentration are favorable sites for the underground reservoir in the study area.

It should be noted that, because of the limitation of the sampling conditions, the number of samples in this study is limited, and seasonal variations were not investigated in this research. The current results are relatively rough. Owing to the simple geology with horizontal/flat strata and no faults and relatively stable monthly mine inflow, we consider that the current results still can reflect the overall distribution characteristics of water quality in the study area. That is, the results are meaningful. On the other hand, the hydrogeological conditions of a coal mine vary dynamically with mining. The results of this study are mainly aimed at the current mining conditions. In the future, when mining the lower/deeper different coal seams, the mine water sources, flow path, and water–rock interaction types may be different and, correspondingly, the mine water quality may vary, which will affect the site selection, construction, and operation of underground reservoirs. Attention should be paid to this dynamic variation in the hydrogeological conditions with coal mining.

## 4. Conclusions

(1) The main over-standard variables are $Na^+$, $F^-$, $SO_4^{2-}$, TDS, and SAR and they are the major concerns for irrigation in the study area. A strong positive correlation exists between $F^-$ and SAR and a negative correlation exists between $F^-$ and $Ca^{2+}$. $Na^+$ concentration in the mine water is affected by the dissolution of rock salt and silicate, as well as reverse cation exchange; $F^-$ concentration is affected by reverse cation exchange and replacement between $OH^-$ in alkaline water and $F^-$ adsorbed on the surface of minerals. Fluorite, rock salt, and gypsum are in an unsaturated state in the mine water and they dissolve and release $Na^+$ and $F^-$ in goaf.

(2) The mine water quality varies with space, primarily because of the fact that the stratigraphic texture on the east of the WL River is different from that on the east of the WL River. On the plane, the mine water quality is better on the east than on the west of the WL River. The water quality of the WL River shows a downward trend in the process of flowing through the study area from north to south. The water quality of the mine water is good in the shallow areas and decreases with the increase in mining depth on the east of the WL River.

(3) The mine water mainly originates from paleo-atmospheric precipitation under a relatively cold and humid paleoclimate and it has stayed in the strata for a long time, which is beneficial to mineral dissolution and cation exchange adsorption.

(4) The goafs with poor mine water quality and adverse water–rock interactions are not suitable for the construction of underground reservoirs. Water–rock interactions in goaf may further increase the concentrations of $F^-$ and $Ca^{2+}$ and SAR in water and are not conducive to improving the water quality. Favorable sites for underground reservoirs lie on the east of the WL River, such as the shallow coal seams at the SGT, DLT, and WL mines, where the mine water has low background values of $Na^+$ and $F^-$ concentrations and SAR. The outcomes of this research can benefit the site selection and construction of an underground reservoir in similar coal mining areas.

**Author Contributions:** Conceptualization, Y.G. and G.L.; methodology, G.L.; software, L.W.; validation, G.L. and Z.Z.; formal analysis, G.L.; investigation, G.L., Z.Z. and L.W.; resources, Y.G.; data curation, L.W. and G.L.; writing—original draft preparation, G.L.; writing—review and editing, Y.G. and L.W.; visualization, Z.Z.; supervision, Y.G.; project administration, Y.G.; funding acquisition, G.L. All authors have read and agreed to the published version of the manuscript.

**Funding:** This research was funded by the Open Fund of State Key Laboratory of Water Resource Protection and Utilization in Coal Mining (Grant Nos. SHJT-17-42.8 and SHJT-17-42.18), the Natural Science Foundation of China (Nos. 42072204 and 41802192), the Key Project of Coal-Based Low-carbon Joint Research Foundation of NSFC and Shanxi Province (No. U1910204), and the Fund of Outstanding Talents in Discipline of China University of Geosciences (Wuhan) (No. 102-162301192664).

**Data Availability Statement:** Datasets analyzed during the present study are accessible from the current author upon reasonable request.

**Conflicts of Interest:** The authors declare no conflict of interest.

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
