# Peer review of "Hydrochemical Characteristics of Mine Water and Their Significance for the Site Selection of an Underground Reservoir in the Shendong Coal Mining Area"

_water, doi:10.3390/w15061038_

Round 1
Reviewer 1 Report
Abstract
The abstract should be rewritten following the basic structure provided by the journal/publisher. Grammatical errors should be corrected, and the conclusion and main results should be at the end of the abstract. The relationship between the ions should be discussed. Remove general information.
Introduction:
Authors are asked to correct language errors and pay attention to the structure and coherence of the introduction section. State the research gap in this paper. Also, the literature review needs to be expanded, especially that on methods CCME-WQI in groundwater, and the results of several previous studies should be included. Details on the advantages and disadvantages of this research in an additional paragraph.
Study area and sampling
The authors are asked to indicate the protocols used for sampling and sample preservation until the samples are treated in the laboratory. What is the reason for selecting these ions? What about the other ions and metals?
Methods
What attempts were made to monitor possible ions in the laboratory environment? Did the authors take blank samples during all experimental procedures? Were deionized water and all chemicals used directly or filtered with filter paper? Please explain in detail. A total of 26 samples were collected, why this sample size? QA/QC should be added. The error percentage of ions should be added.
Results and discussion
Your relationship to point and non-point sources. What about seasonal variations? Link to CCME-WQI. The discussions are not comprehensive and mature, and the authors have limited themselves only to general statements and conclusions in each section. The conceptual model of this study is necessary for water resources detection and management. What about the concentration of the other ions? Stronger discussion of the section on enrichment.
Conclusion
The conclusion needs to be restated with the key features and management tactics or recommendations of this study.
Reviewer 2 Report
The article is devoted to solving an important applied problem: optimizing the location of underground water reservoirs in coal-mining water-deficient areas, taking into account the hydrochemical characteristics of mine waters. Despite the regional nature of the study, the results obtained can be extended to other regions with similar water use problems. The reviewer has no fundamental comments on the submitted materials, however, the work is framed very carelessly and before publication it is required to bring it into line with the level of the Journal. In particular, the following changes are recommended. 1) On figures with small captions (maps in Figures 1 and 2, axes labels in Figures 5, 6, and 7), enlarge the font to a well-readable one and bring the design of all graphs to a single standard. 2) Numerical values in the text and tables should preferably be rounded to three significant figures (for example, 12.73~951.71 mg/L should be replaced by 12.7~952 mg/L). 3) Symbols of chemical elements and SAR everywhere, including in formula (1) and in the text before and after it, should be indicated in roman type instead of italics. 4) On the lines following the formulas, replace “Where” with paragraph indentation with “where” without paragraph indentation. 5) Correct the quality of the text presentation and the list of references (a lot of double spaces or, conversely, their absence in the right places, as well as other typos, for example “Table 1. classification …”, “Table 2. statistics …”.
Round 2
Reviewer 1 Report
Authors revised as per the comments now it can be accepted for publication